# New Insights into the Biological Response Triggered by Dextran-Coated Maghemite Nanoparticles in Pancreatic Cancer Cells and Their Potential for Theranostic Applications

**DOI:** 10.3390/ijms24043307

**Published:** 2023-02-07

**Authors:** Mihaela Balas, Daniela Predoi, Carmen Burtea, Anca Dinischiotu

**Affiliations:** 1Department of Biochemistry and Molecular Biology, Faculty of Biology, University of Bucharest, 91-95 Splaiul Independentei, 050095 Bucharest, Romania; 2National Institute of Materials Physics, 076900 Magurele, Romania; 3Department of General, Organic and Biomedical Chemistry, NMR and Molecular Imaging Laboratory, University of Mons, Avenue Maistriau 19, Mendeleïev Building, B-7000 Mons, Belgium

**Keywords:** iron oxide nanoparticles, PANC-1 cells, nanotoxicity, theranostic, caspase-1, Hsp60, Hsp90, p53

## Abstract

Iron oxide nanoparticles are one of the most promising tools for theranostic applications of pancreatic cancer due to their unique physicochemical and magnetic properties making them suitable for both diagnosis and therapy. Thus, our study aimed to characterize the properties of dextran-coated iron oxide nanoparticles (DIO-NPs) of maghemite (γ-Fe_2_O_3_) type synthesized by co-precipitation and to investigate their effects (low-dose versus high-dose) on pancreatic cancer cells focusing on NP cellular uptake, MR contrast, and toxicological profile. This paper also addressed the modulation of heat shock proteins (HSPs) and p53 protein expression as well as the potential of DIO-NPs for theranostic purposes. DIO-NPs were characterized by X-ray diffraction (XRD), transmission electron microscopy (TEM), dynamic light scattering analyses (DLS), and zeta potential. Pancreatic cancer cells (PANC-1 cell line) were exposed to different doses of dextran-coated ɣ-Fe_2_O_3_ NPs (14, 28, 42, 56 μg/mL) for up to 72 h. The results revealed that DIO-NPs with a hydrodynamic diameter of 16.3 nm produce a significant negative contrast using a 7 T MRI scanner correlated with dose-dependent cellular iron uptake and toxicity levels. We showed that DIO-NPs are biocompatible up to a concentration of 28 μg/mL (low-dose), while exposure to a concentration of 56 μg/mL (high-dose) caused a reduction in PANC-1 cell viability to 50% after 72 h by inducing reactive oxygen species (ROS) production, reduced glutathione (GSH) depletion, lipid peroxidation, enhancement of caspase-1 activity, and LDH release. An alteration in Hsp70 and Hsp90 protein expression was also observed. At low doses, these findings provide evidence that DIO-NPs could act as safe platforms in drug delivery, as well as antitumoral and imaging agents for theranostic uses in pancreatic cancer.

## 1. Introduction

Pancreatic cancer is one of the most challenging forms of cancer regarding treatment, as the majority of the patients do not respond to currently available treatments. Despite the promising increase in the survival rate from 5% to 11% reported by the American Cancer Society in 2022, it is predicted that pancreatic cancer will become the second leading cause of cancer-related deaths before 2030. The main limitations in treating patients with this disease are: (i) the lack of detection tools for its diagnosis in the early stages and (ii) the aggressiveness of the tumors with multiple levels of therapeutic resistance [1].

Imaging is very important in the management of pancreatic cancer, the most important methods being computed tomography (CT), magnetic resonance imaging (MRI), and endoscopic ultrasonography (EUS). Even though EUS is the most effective technique in the detection of this disease, it has some limitations in identifying extra-abdominal metastasis, which are observed by performing CT or MRI [2].

The problem for patients with pancreatic cancer is the late detection, often accompanied by distant metastasis at the time of diagnosis [3]. Even radical surgery that is applicable to 10–20% of pancreatic cancer patients is not sufficient for a cure [4] with other therapies being necessary. Despite novel targeted strategies and antitumor immunotherapy used in the cure of malignant tumors, for pancreatic cancer, chemotherapy remains the most important therapeutic approach [5]. However, the clinical studies revealed that pancreatic tumors become chemo-resistant due to cellular intrinsic and extrinsic factors related to the tumor microenvironment [6,7]. Therefore, a theranostic approach that combines effective treatment with imaging monitorization is highly desirable to improve the survival of patients with pancreatic cancer.

Nanotechnology offers great possibilities for designing platforms for cancer theranostics as it can provide the co-delivery of multiple types of therapeutic drugs and imaging agents and their accumulation in tumors [8]. Different nanoparticles (NPs) were previously used to develop nanosystems for theranostic applications of pancreatic cancer [9].

Iron oxide nanoparticles (IONPs) are among the most promising candidates due to their unique physicochemical and magnetic properties being suitable for both diagnosis and therapy [10]. These NPs can combine the capability to enhance contrast in magnetic resonance imaging (MRI) with the ability to treat pancreatic cancer through multiple emerging procedures such as hyperthermia [11], the active or passive delivery of drugs, proteins, antibodies, and nucleic acids [12,13,14,15], magnetic drug targeting, targeted immuno-activation [16], and, potentially, ferroptosis [17].

IONPs including magnetite (Fe_3_O_4_) or maghemite (γ-Fe_2_O_3_) have been extensively studied and widely used as MRI contrast agents in clinical diagnosis due to their ability to shorten T2 relaxation times. In comparison with conventional gadolinium-containing T1 contrast agents, IONPs can be magnetically saturated in the normal range of magnetic field strengths used in MRI scanners (usually 3 Tesla) resulting in a greater negative contrast in T2-weighted images [18]. The IONPs used in MRI are called superparamagnetic iron oxide nanoparticles (SPIONs) and are characterized by their small size (< 100 nm) and superparamagnetism, which is the ability of the NPs to flip the orientation of the magnetic moments within their core rapidly when exposed to an external magnetic field. This gives SPIONs very high relaxivity values that generate image contrast of the surrounding soft tissue where particles accumulate [19].

Since bare SPIONs tend to aggregate by dipole–dipole interactions, surface coating is required to enhance the colloidal stability and dispersibility of particles, thus increasing their blood circulation time and cellular uptake, and minimizing non-specific interactions in order to reduce toxicity and make particles biocompatible. Moreover, the coating protects the NP surface from oxidation, and provides chemical groups for the conjugation of drug molecules and targeting ligands [20]. Coatings can be fabricated from different polymers such as dextran, chitosan, starch, gelatine, poly(lactic-co-glycolic acid) PLGA, and polyethylene glycol (PEG), as well as silica, liposomes, albumin, etc. [21].

Dextran is one of the most common coatings and can be covalently cross-linked to compounds with amine groups ready for conjugation to drugs and ligands. Dextran-coated SPIONs are a well-established platform for diagnostic imaging by MRI and some candidates (e.g., Ferumoxytol, Resovist) of this class have been approved by the Food and Drug Administration of the USA for their medical use as contrast agents in MRI of liver tumors [22]. However, some formulations such as Sinerem, Combidex, and Feridex have been withdrawn from clinical use due to adverse reactions that occurred in patients [23].

Upon coating, the toxic effects of SPIONs may still exist, and for this reason, they should be evaluated with caution. The toxicity of SPIONs may depend on various factors such as size, shape, structure, solubility, concentration, surface modification, and cell/tissue type [24,25,26].

Generally, in vivo studies showed a low toxicity of dextran-coated SPIONs, but damaging effects were also reported [27,28] as well as activation of the kallikrein–kinin system [29] or chronic iron toxicity due to the administration of high doses of particles [26].

In cells, dextran-coated SPIONs locate in lysosomes or endosomes and are released in the cytoplasm following enzymatic degradation into free iron ions or disruption of the proton pump balance that leads to a lysosomal rupture and iron release. Increases in cellular iron levels enhance the reactive oxygen species (ROS) production via the Fenton reaction leading to oxidative stress and further to lipid peroxidation, protein and DNA damage, and cell death. Furthermore, ROS can perturb actin cytoskeleton modulation, gene expression, redox regulation, or cellular signaling [26]. Depending on their type, cells use different pathways to defend themselves against SPIONs, and thus ROS levels may vary.

In pancreatic cancer cells, dextran-coated SPIONs exhibited low cytotoxicity at doses below 60 μM [30]. Since many proposed dextran-coated SPIONs may be potentially harmful, a complete toxicological profile is mandatory prior to their intended usage to establish the toxicity thresholds, kinetics, and toxicity mechanisms of these particles and thus minimize potential health hazards upon their exposure.

In this context, our aim was to evaluate the effects of dextran-coated iron oxide NPs (DIO-NPs) of maghemite (γ-Fe_2_O_3_) type on pancreatic cancer cells at a low dose versus a high dose and their potential for theranostic applications. This is a preliminary study focused on the MRI contrast, cellular uptake, and toxicological profile of these NPs in vitro. Aspects regarding the cytotoxic dosage, cell iron loading, induction of oxidative stress, and cell death, as well as the modulation of heat shock proteins (HSPs) and p53 expression, were particularly addressed in this paper.

## 2. Results

### 2.1. Characterization of DIO-NPs

The coated maghemite nanoparticles show improved dispersibility, which leads to better stability [31,32]. The dextran used as a surfactant in this study protects the maghemite core against various oxidation incidents. At the same time, the dextran on the surface can provide anchor points for different agents.

The phase purity and structure of the synthesized DIO-NP were investigated by XRD (Figure 1a). Scherrer’s formula (D = (0.9∙λ)/(βcosθ)) was used to calculate the average size of the crystallites from the broadening of the XRD line [33]. The diffraction peaks of the DIO-NPs detected were assigned to the Miller indices values (hkl) of (220), (311), (400), (422), (511), and (440). The observed diffraction peaks were indexed to a face-centered cubic spinel structure (Fd3m3 space group). All identified diffraction peaks correspond to maghemite with cubic structure ICSD-PDF No. 1346. It is well known that the identification of magnetite and maghemite phases is difficult to quantify because they are very close [34,35]. However, after calculating the lattice parameter, we were able to say whether the identified phase was maghemite or magnetite. For the synthesized DIO-NP, the lattice parameter was 8.35 Å. The calculated value for the lattice parameter was equal to that of the bulk lattice parameter of γ-Fe3O4 (a = 8.3515 Å), which shows that the identified phase was that of maghemite [36]. The average crystallite size of the DIO-NPs calculated from the XRD data was 8.2 ± 1 nm. The XRD patterns of the DIO-NPs certified their crystallinity. No other phases except the maghemite were detectable. The morphology and particle sizes of the DIO-NPs were determined by TEM analysis. Figure 1b reveals that the DIO-NPs had spherical shapes. Moreover, no agglomeration of the nanoparticles was observed.

The DIO-NP particle size distribution histogram was conducted after measuring the mean diameter (DTEM) of approximately 500 particles. The average size of the DIO-NPs was 8.9 ± 1.5 nm (Figure 1c). The sizes of the particles counted to obtain the distribution were between 4 and 14 nm. The selected area electron diffraction (SAED) pattern is presented in Figure 1d. The SAED pattern of the synthesized DIO-NP presented in Figure 1d was indexed by a cubic maghemite (PCPDF#872334). The diffraction rings revealed were attributed to the (220), (311), (400), (422), (511), and (440) planes. The rings observed were in good accordance with the XRD pattern.

In order to investigate the effect of dextran on the size of the analyzed sample, DLS experiments were conducted. DLS measurements revealed that the hydrodynamic diameter (DH) of the DIO-NP was 16.3 ± 1.52 nm (Figure 1e). This result suggests that the presence of dextran results in an increase in the size of the synthesized sample, which is in agreement with previous studies [37,38,39]. Comparing the crystallite sizes as evaluated by TEM and XRD analysis (DTEM/DXRD = 1.08), it is observed that there is no significant difference. On the other hand, DH/DXRD = 1.99, which suggests that dextran is present on the surface of maghemite nanoparticles.

### 2.2. Cytotoxicity of DIO-NPs on PANC-1 Cells

The viability assay showed that the metabolic activity of PANC-1 cells was affected in a concentration- and time-dependent manner (Figure 2). The cytotoxicity of the DIO-NPs was observed starting with 48 h of incubation and a dose of 42 μg/mL. After 72 h, the toxic effect of the DIO-NPs became stronger, and PANC-1 cell viability decreased by almost 50% at a concentration of 56 μg/mL (1 mM) compared to the control (*** *p* < 0.001). In cells treated with a concentration less than 28 μg/mL (0.5 mM), no significant cytotoxicity was registered during the tested exposure intervals. For the following determinations, two doses of DIO-NPs (28 and 56 μg/mL) were chosen.

### 2.3. Cellular Uptake of DIO-NPs

DAB-enhanced Prussian blue stain (Figure 3a) showed a dose and time-dependent accumulation of iron deposits (brown color) in the cytoplasm of PANC-1 cells (light blue color) after incubation with DIO-NPs. Exposure to the non-toxic concentration of DIO-NPs (28 μg/mL) led to a slight iron uptake in PANC-1 cells after 48 h, but the number of iron clusters visibly increased after 72 h, being localized mostly around the nucleus and possibly inside endocytic vesicles such as lysosomes. Moreover, in cells incubated with the cytotoxic concentration of DIO-NPs (56 μg/mL), iron loading was observed starting with 24 h of exposure and increased to 72 h when almost the entire cell cytoplasm and the nucleus appeared to be covered in brown iron deposits. Additionally, a decrease in cell density was observed after incubation with 56 μg/mL of DIO-NPs beginning from 48 h, thus confirming the MTT assay results.

To quantify the amount of DIO-NPs uptaken by pancreatic cancer cells, two different techniques were used: colorimetry and relaxometry. Thus, it was noted that the cellular iron uptake increased in time proportionally with the applied concentration of DIO-NPs. The colorimetric Perl’s reaction (Figure 3b) showed that after incubation for 48 and 72 h of PANC-1 cells with DIO-NPs at a concentration of 28 μg/mL, the relative intracellular iron level increased, respectively, by 9% (* *p* < 0.05) and 40% (** *p* < 0.01). At the same time intervals, for DIO-NPs at a concentration of 56 μg/mL, increases in total cellular iron by 77% (*** *p* < 0.001) and 154% (*** *p* < 0.001), respectively, were found.

The quantification of iron by relaxometry showed similar results (Figure 3c). After incubation for 24, 48, and 72 h with 28 μg/mL of DIO-NPs, the means of the iron content in PANC-1 cells were 14, 18, and 37 picograms/cells, respectively, while in cells incubated with 56 μg/mL DIO-NPs, they were 21, 32, and 46 picograms/cell.

### 2.4. Biogenesis and Subcellular Localization of Lysosomes

Compared with the control, the number of lysosomes in the cells incubated with DIO-NPs was higher and dependent on the exposure interval and NP concentration (Figure 2d,e). After the incubation of PANC-1 cells with DIO-NPs at a dose of 28 μg/mL, a significant increase in the density of lysosomes was observed only after 48 h (by 47%, * *p* < 0.05) of exposure and become more intense after 72 h (by 158%, *** *p* < 0.001). In cells exposed to a dose of 56 μg/mL, a gradual increase in the lysosomes’ number was noticed from 24 h to 72 h of incubation when the fluorescence intensity of LysoTracker Green DND-26 reached 168% (** *p* < 0.01) more than the control. Figure 2e shows the subcellular localization of lysosomes, which are dispersed in the entire cytoplasm, predominantly on the edge of the cells. Some clusters of lysosomal vesicles were formed in the cytosol of PANC-1 cells around the nucleus, most likely resulting from endocytosis of the DIO-NPs.

### 2.5. MR Contrast

Figure 4 shows the imaging contrast (T2-weighted imaging) of samples containing PANC-1 cells exposed to DIO-NPs at concentrations of 0, 28, and 56 μg/mL iron for 24, 48, and 72 h. To measure T2, a multi-slice multi-echo sequence was used with a repetition time (TR) of 4000 ms. The echo time (TE) was increased in 15 equal steps from 20 ms to 320 ms for 24 h and 48 h time exposure intervals (Figure 4b,c), and from 10.8 to 172.57 for 72 h. The MRI images shown in Figure 4a were recorded at TE of 140, 140, and 10.8 ms, respectively. The enhanced cellular uptake of DIO-NPs (at both doses) resulted in the darkening effect of MR images compared to the control cells. The strongest T2 contrast (dark contrast) was obtained in cells incubated with the highest dose of DIO-NPs (56 μg/mL) for 72 h (Figure 4d). The MRI contrast of the DIO-NPs in PANC-1 cells was determined by curve fitting of the signal intensity decay.

### 2.6. Potential Toxic Effects of DIO-NPs

The toxicity of the DIO-NPs was assessed by measuring the level of oxidative stress-related biomarkers (ROS production, GSH content, and MDA level), LDH release from necrotic cells, and caspase-1 enzymatic activity as an inflammatory response.

According to Figure 5a, the production of ROS in PANC-1 cells increased in a dose- and time-dependent manner. Starting with 24 h, a significant elevation in the intracellular ROS level was registered in cells treated with both concentrations of DIO-NPs. After 72 h of incubation using the dose of 56 μg/mL, the ROS level increased by three times (*** *p* < 0.001) compared to the level found in the control cells.

Furthermore, it was noticed that the GSH content was not significantly altered in cells incubated with the 28 μg/mL DIO-NPs during the experiment (Figure 5b). In contrast, in cells exposed to a double dose of NPs, the GSH pool was significantly depleted (by 35% ** *p* < 0.01) starting at 24 h. After 48 and 72 h, a dramatic reduction of 83% and 92%, respectively, was noticed compared to the control.

Lipid peroxidation, which reflects the oxidative degradation of polyunsaturated fatty acids, was measured by MDA levels. As illustrated in Figure 5c, the significant changes appeared only after 48 and 72 h in cells incubated with DIO-NPs at a concentration of 56 μg/mL when the MDA level increased by 45% (** *p* < 0.01) and 146% (*** *p* < 0.001), respectively.

LDH activity in the culture medium of cells treated with NPs indicated cell membrane damage as a sign of necrotic death. As shown in Figure 5d, a significant elevation of LDH activity was detected after 48 and 72 h of exposure to DIO-NPs at a concentration of 56 μg/mL by 27% (*** *p* < 0.001) and 63% (*** *p* < 0.001), respectively. In cells treated with the 28 μg/mL DIO-NPs, LDH activity did not exceed the control level.

Similarly, an amplification of the caspase-1 activity was registered only in cells treated with the higher dose (Figure 5e). The caspase-1 activation known to initiate the inflammatory programmed cell death called pyroptosis was achieved in PANC-1 cells starting with 24 h of exposure to the DIO-NPs when caspase-1 reached the highest activity level (by 29% compared to control, *** *p* < 0.001). After 48 and 72 h, the level of caspase-1 activity was maintained over that of the control, but a slight reduction in this enzymatic activity, dependent on exposure intervals, was also noticed.

### 2.7. Alterations in Protein Expression of Heat Shock Proteins and Tumor Suppressor p53

Heat shock proteins (HSPs) are a group of molecular chaperones with different functions that protect the integrity of the structure of cellular proteins in response to stress conditions. To track the modifications in the overall status of the proteome, the protein expression of some members of the HSP family (Hsp90, Hsp70, and Hsp60) was analyzed in PANC-1 cells exposed to DIO-NPs (Figure 6a–d). Thus, it was found that the level of the Hsp90 protein increased significantly by 38% (** *p* < 0.01) in the first 24 h in cells treated with a 56 μg/mL DIO-NP dose, and then it was restored to the control level up to 72 h. When a dose of 28 μg/mL of the DIO-NPs was applied, an upregulation of the Hsp90 expression was registered only after 48 h (by 29%, * *p* < 0.05). The Hsp70 protein expression was inhibited after 24 h by 35% (*** *p* < 0.001) in cells exposed to 28 μg/mL of the DIO-NPs and by 45% (*** *p* < 0.001) in cells exposed to 56 μg/mL of the DIO-NPs compared to the control.

After 48 h of exposure, the level of Hsp70 was still lower than the control but the result was statistically significant only for the cells treated with the smaller dose. Next, a dose-dependent upregulation of Hsp70 expression was noticed after 72 h of exposure to 28 (49%, ** *p* < 0.01) and 56 μg/mL of DIO-NPs (62%, * *p* < 0.05).

No modification of the Hsp60 protein expression was observed in any of the conditions except for the cells treated with 28 μg/mL of DIO-NPs for 72 h when a significant elevation in the Hsp60 level by 39% compared to the control was noticed.

The protein expression of the tumoral suppressor p53 was also evaluated to anticipate the mechanism of the antitumor effects of DIO-NPs. Yet, in our study, the exposure of PANC-1 cells to DIO-NPs had no significant effect on p53 protein expression (Figure 6a,e).

## 3. Discussion

In this study, we found that dextran-coated ɣ-Fe_2_O_3_ NPs with a hydrodynamic diameter of 16.3 nm produce a significant negative contrast using a 7 T MRI scanner following a substantial cellular iron uptake and suppress the viability of PANC-1 cancer cells to 50% at a concentration of 56 μg/mL by inducing oxidative stress and necrotic cell death. Up to a concentration of 28 μg/mL, DIO-NPs were well tolerated by cells, showing high biocompatibility. According to a previous study [34], the agglomeration of magnetic particles is due to the presence of high surface energy between the prepared magnetic NPs and the presence of magnetic dipole–dipole interactions.

The core of maghemite was covered with dextran to improve its dispersibility and reduce its toxic effects. Different techniques were used to determine the size of the DIO-NPs, knowing that their size plays the most important role in the success of their applications. Following DLS studies, the hydrodynamic diameter of the DIO-NPs was determined. Moreover, the presence of dextran on the surface of the magnetic core was highlighted. The colloidal stability of the DIO-NP suspension was indicated by the zeta potential value of −31.7 mV. NPs’ stability is very important for biomedical applications to achieve the expected and consistent outcomes [31,40,41]. The low value of the zeta potential of MNPs implies that the NP can show poor stability in aqueous solutions [32,42]. In agreement with previous studies [41,42], low zeta potential values improve Van der Waals interparticle attractions and cause the rapid coagulation and flocculation of nanoparticles. On the other hand, NPs can show good stability in aqueous solutions at higher values of the zeta potential. Very recent studies have shown that there is a specific value of the zeta potential (≈±30 mV) that determines the stability of NPs. At this value, high electrostatic repulsion forces appear between the NPs [32,42,43].

To understand the biological behavior of DIO-NPs and their triggered cellular responses, in vitro experiments were performed on a PANC-1 cell line. This cell line derived from a human pancreatic ductal adenocarcinoma is frequently used as an in vitro model in pancreatic cancer research and the evaluation of novel nanotherapeutics because of its morphological (epithelial) and genetic characteristics [44,45]. Herein, PANC-1 cells were exposed to different doses of DIO-NPs (between 14–56 µg/mL) for 24, 48, and 72 h to assess the cellular uptake of NPs and their MRI performance as contrast agents as well as the potential cytotoxic effects and molecular modifications after cell–particles interaction.

A cell viability assay revealed that the toxicity of our nanoformulation of DIO-NPs starts with a dose of 42 μg/mL reaching a 50% reduction in cell metabolic activity at a dose of 56 μg/mL (1 mM) of DIO-NPs after 72 h. Different from our results, Lafunete-Gomez et al. (2021) [46] indicated that dextran-coated ɣ-Fe_2_O_3_ NPs with a larger size (of around 100 nm) caused no toxicity in PANC-1 cells up to a concentration of 2 mg Fe/mL after 72 h. Another study showed that, without affecting cell viability, dextran-coated ɣ-Fe_2_O_3_ NPs of about 75 nm cause oxidative stress and genotoxicity at a dose of 4 μg/mL after 24 h [47]. In this regard, further research will be necessary to determine an exact time frame for the clearance of DIO-NPs by the reticuloendothelial system (RES) in vivo in order to prove their cytotoxicity on pancreatic cells.

Thus far, maghemite NPs are considered cytocompatible being one of the most preferred materials for IO-based nanosystems because of the iron (III) ions in their composition that are already found in the human body [48]. However, data from the literature regarding the toxicity of these NPs are mixed. Different cellular responses in vitro and in vivo are a consequence of the multiple factors that may influence the toxicity of these NPs such as: dose, exposure time, particle size, colloidal stability, surface properties, cell type, cellular uptake, etc. [49].

We suppose that the size of DIO-NPs and their high dispersity in the suspension is the cause of cytotoxicity displayed over a concentration of 42 μg/mL. This assumption is supported by the high cellular internalization of DIO-NPs observed in PANC-1 cells. Particles of small sizes can be uptaken more easily by cells, and iron accumulates in the cytoplasm in a higher amount, thus leading to a strong toxicity [50]. In our study, a significant cellular uptake of DIO-NPs occurred after 48 h in pancreatic cancer cells dependent on dose and exposure time. Visible deposits of iron brown stains were observed in cells around the nucleus at a dose of 28 μg/mL while at a higher dose (56 μg/mL), a massive iron loading was obtained after 72 h where iron was dispersed in the entire cytoplasm. The intracellular accumulation of iron was also confirmed by both relaxometry and spectrophotometric measurements.

Iron internalization is usually related to higher toxicity levels and may be modulated by various factors including NP size, surface coating, media serum concentration, medium components, and iron redox state [37]. Moreover, Luengo et al. (2013) [51] found that neutrally charged dextran-coated maghemite NPs were less internalized by L929 and Saos-2 cells compared to surface-charged maghemite NPs coated with dimercaptosuccinic acid (DMSA) or (3-aminopropyl)triethoxysilane (APS). The study indicated that the absence of the surface charge prevented the adsorption of cell media components and the formation of the protein corona on dextran-coated NPs, conditions that seemed to promote the cellular uptake of surface-charged NPs and increased their cytotoxicity. Our data showed that the presence of serum proteins in the cell culture medium affected the electrostatic potential of DIO-NPs as the zeta potential increased up to −7.5 mV after 72 h [52], suggesting a significant change in the surface charge of the NPs.

Furthermore, we investigated the level of lysosomes in PANC-1 cells after DIO-NP exposure. Fluorescent labeling of lysosomes showed a strong increase in the number of lysosomes dependent on dose and exposure time, which suggests the internalization of NPs in the pancreatic cancer cells. MRI analyses also confirmed this, showing efficient labeling dependent on dose and exposure time, enabling a high contrast on T2-weighted images. The accumulation of NPs in the cells is an essential parameter to improve the MRI resolution that has to be high enough to enhance the contrast of cells in the deep tissues while maintaining the NP dosage within the biocompatibility limits. A high accumulation of iron in the cytoplasm leads to increased levels of ROS as we showed for the PANC-1 cells exposed to the 56 μg/mL DIO-NPs. However, the cells exposed to the 28 μg/mL DIO-NPs induced a moderate time-dependent production of ROS that did not cause any other alterations of biological molecules.

As shown before, IO-NPs may generate ROS in the cytoplasm via the Fenton reaction. After internalization, NPs are disintegrated into the acidic microenvironment of endolysosomes, and the Fe^2+^/Fe^3+^ ions released in the cytoplasm can react with hydrogen peroxide and lead to highly reactive hydroxyl [53]. Moderate levels of ROS can be neutralized by cancer cells as they can adapt to develop powerful antioxidant mechanisms to counteract oxidative stress [54].

However, excessive ROS production leads to oxidative stress, which can further damage lipids, proteins, cofactors of enzymes, and DNA, thus inducing cell death of tumor cells. In this context, the increased production of ROS in pancreatic cancer cells exposed to 56 μg/mL of DIO-NPs correlated well with depletion of GSH content, increased levels of MDA, leakage of LDH outside the cells, and elevation of caspase-1 activity. Here, we showed that the high production of ROS overwhelmed the antioxidant capacity of PANC-1 cells by dramatically reducing the level of GSH, the major cellular thiol protein that participates in antioxidant defense, and induced lipid peroxidation starting with 48 h of exposure.

Furthermore, the amount of LDH released in the culture medium increased significantly after 48 h of exposure to DIO-NPs, indicating the loss of PANC-1 cell membrane integrity and, at the same time, the occurrence of necrotic cell death triggered probably by an enhanced lysosomal degradation. Interestingly, the level of caspase-1 activity increased after 24 h of exposure, but it started to slightly decrease up to 72 h. Caspase-1 is a protease that along with NLPR3 protein and the adaptor ASC are critical components of the inflammasome protein complex that participates in the inflammatory response against harmful stimuli by cleaving the precursors of the proinflammatory cytokines IL-1β and IL-18. Moreover, it activates pyroptosis, a programmed lytic cell death, through cleavage of gasdermin D [55]. Thus, we show that as well as oxidative stress, an inflammatory response of PANC-1 cells exposed to 56 μg/mL of DIO-NPs was putatively initiated in the first 24 h. The decreasing tendency of caspase-1 activity over time suggests inhibition of the inflammation process that might be a consequence of the altered redox homeostasis, which can modulate the activation potential of the NLRP3 inflammasome [56].

To gain a deeper understanding of their biological behavior in pancreatic cancer cells, we analyzed the effects of DIO-NPs on the protein expression of Hsp90, Hsp70, and Hsp60 as well as of p53. HSPs are a family of proteins of which expression on the transcriptional level is mainly induced by heat shock factor 1 (Hsf1), a master regulator of cells’ response to stressful conditions. They maintain protein homeostasis and can interact with multiple critical components of signaling pathways that regulate cell growth and development. Hsp90 particularly is a major therapeutic target for cancer because it protects numerous oncoproteins [57]. The significant increase in the Hsp90 protein level in the first 24 h of exposure to a concentration of 56 μg/mL of DIO-NPs might be the cause of a protective response of pancreatic cancer cells to stress induction. However, after 48 h, the protein level of Hsp90 returned to normal. Interestingly, exposure to a concentration of 28 μg/mL of DIO-NPs produced an upregulation of Hsp90 protein expression only after 48 h, which, most likely, is a defense mechanism of pancreatic cancer cells to ROS induction.

The major significant changes in HSP expression registered after DIO-NP exposure were produced for the Hsp70 protein. In this case, both doses caused the same alterations, which started with a significant downregulation of the Hsp70 level after 24 h followed by a substantial upregulation of its protein expression up to 72 h. Hsp70 is rapidly induced to protect cells in stressful conditions by helping proteins to maintain their native conformations. However, Hsp70 also regulates various components of signaling pathways facilitating the conformational changes required for their activation, it can suppress the release of other proinflammatory factor molecules, and it plays an antiapoptotic role [58]. In a recent study, it was shown that inhibition of Hsp70 leads to the activation of the proteasomal system in fibroblasts to maintain the proteome balance of the cell by avoiding protein oxidation and aggregation [59]. In this context, the depletion of Hsp70 expression in PANC-1 after 24 h of exposure to DIO-NPs might be a result of activation of the cell protection mechanism against mild oxidative stress, and the enhancement of Hsp70 expression is correlated with the stronger oxidative environment induced after a longer exposure period.

Surprisingly the expression of Hsp60 increased only in PANC-1 cells exposed for 72 h to 28 μg/mL of DIO-NPs, which indicates a low contribution of this protein to the cellular response against the cytotoxic effects of these NPs. Still, it was revealed that enhanced Hsp60 expression promoted the growth of pancreatic ductal cancer cells [60]. Compared with the other two HSPs located in the cytoplasm, Hsp60 is mainly found in the mitochondria which might limit the interaction with NPs or other stressful stimuli.

The expression of tumor suppressor protein p53 in pancreatic cancer cells was not significantly modified in the conditions of our study, which means the mechanisms of the toxic effects induced by DIO-NPs are probably independent of p53 regulation.

This study revealed that synthesized DIO-NPs with a hydrodynamic diameter of 16.3 nm accumulate in a higher proportion in PANC-1 cells, thus providing an enhancement of MR contrast. Moreover, we unveil the cellular response of pancreatic cancer cells after exposure to non-cytotoxic and cytotoxic doses of DIO-NPs for up to 72 h. As a toxicity mechanism in addition to oxidative stress, these NPs could initiate an inflammatory response in PANC-1 cells at a dose of 56 μg/mL in the first 24 h.

DIO-NPs are biocompatible in vitro at doses below 28 μg/mL, which allows their utilization as contrast agents in MRI while loaded with therapeutic agents. Yet, the toxicity of DIO-NPs at higher doses might be advantageous if exploited for the eradication of cancer cells in theranostic applications. Delivering cytotoxic ROS directly to the tumor, or alternatively inhibiting the antioxidant enzyme system, is a strategy that has been suggested previously [61]. However, the complexity of pancreatic cancer cells may limit the accomplishment of the therapeutic purpose of these NPs, and for this reason, more studies are required to understand the in-depth mechanisms and adaptive cellular responses both in vitro and in vivo.

## 4. Materials and Methods

### 4.1. Synthesis and Characterization of Dextran-Coated ɣ-Fe_2_O_3_ Nanoparticles

Dextran-coated maghemite nanoparticles (DIO-NPs) were obtained by an adapted co-precipitation method according to previous studies [62,63,64,65,66]. The synthesis of maghemite NPs coated with dextran (DIO-NPs) was carried out at room temperature and the ratio of Fe2+/Fe3+ was 0.5. The ferric chloride hexahydrate (FeCl_3_ × 6H_2_O) and ferrous chloride tetrahydrate (FeCl_2_ × 4H_2_O) in 2 M HCl were mixed and together with the solution of NaNO_3_ (1 mol/L^−1^) were dropped into a NaOH (2 mol⋅L^−1^) solution under continuous stirring. The black precipitate that formed (magnetite) was transformed into maghemite by repeated treatments with HClO_4_ solution (3 mol⋅L^−1^). The treatment was repeated until the Fe^2+^/Fe^3+^ ratio in the global sample reached 0.05, approximately. After the last treatment, the aqueous suspension of DIO-NPs was centrifuged (10,000 rpm). The resulting precipitate was washed with double distilled water and dispersed in deionized water. This procedure was performed several times. The resulting magnetic fluid was mixed with a dextran solution (10 g in 100 mL of water). The mass ratio of maghemite to dextran was 1:2. The resulting solution of DIO-NPs was washed by means of magnetic columns before being analyzed. The phase purity and the structure of the sample were evaluated by X-ray diffraction measurements (Bruker D8 Advance diffractometer, Cu Kα, λ = 0.15406 nm) with a high-efficiency one-dimensional detector (Lynx Eye type, Karlsruhe, Germany) at 2θ range 20–70°. The step was 0.02°/min and 34 s measuring time per step. Transmission electron microscopy (TEM) was used to assess the morphology (JEOL 200 CX, Jeol, Tokyo, Japan). The mean hydrodynamic diameter and zeta potential of DIO-NP was determined by dynamic light scattering (DLS) using a ZetaSizer Nano ZS (Malvern Instruments Limited, Cambridge, UK). All measurements were effectuated three times at 25 °C.

### 4.2. Preparation of NP Suspension

An amount of 100 mg DIO-NPs was dispersed in 10 mL MilliQ water, on ice, with a UP200S ultrasonicator (Hielscher Ultrasound Technology, Teltow, Germany) set at 40% amplitude and 0.5 cycles. The suspension was centrifuged for 10 min at 5000 rpm and the resulting supernatant was filtered through a polyvinylidene difluoride (PVDF) syringe filter (Millipore, Darmstadt, Germany) of 0.45 µm pore size. Upon filtration, the NP suspension was sterilized using UVC radiation and stored at 4 °C.

### 4.3. Measurement of Relaxivity (r1 and r2)

Relaxivity values (r_1_ and r_2_) are characteristic of the contrast agent efficacy and are expressed in s^−1^ × mM^−1^. They were calculated using following formulas: r_1_ = (R_1_ − R_0_)/C and r_2_ = (R_2_ − R_0_)/C, where R_0_ = relaxation rate in the absence of nanoparticles and C = iron concentration (mM). The longitudinal (R_1_) and transverse (R_2_) relaxation rates (s^−1^) of nanoparticles are defined as the reciprocal of the relaxation time T_1_ (R_1_ = 1/T_1_) and T_2_ (R_2_ = 1/T_2_). T_1_ and T_2_ relaxation times were determined using a Bruker mq60 MR relaxometer (Bruker, Karlsruhe, Germany) operating at 37 °C and 1.4 T (60 MHz).

### 4.4. Cell Culture and Treatment Conditions

The human PANC-1 cell line derived from a pancreatic carcinoma of ductal origin was purchased from American Type Culture Collection (ATCC, CRL-1469, Manassas, VA, USA). The cells were grown in Dulbecco’s Modified Eagle Medium (DMEM, 31600-083, Gibco, Abingdon, UK) supplemented with 3.5 g/L glucose, 1.5 g/L NaHCO_3_, 10% fetal bovine serum (10270-106, origin South America, Gibco, Life Technologies, Carlsbad, CA, USA), and 1% antibiotic-antimycotic solution (A5955, Sigma, St. Louis, MO, USA). The cells were maintained at 37 °C in a humidified atmosphere with 5% CO_2_. For treatment, the cells were seeded at a density of 10^5^ cells/mL and the next day were exposed to different concentrations of DIO-NPs (expressed as µg Fe/mL) of 14, 28, 42, and 56 µg/mL (corresponding to 0.25, 0.5, 0.75, and 1 mM) and incubated for 24, 48, and 72 h. Untreated PANC-1 cells were used as control.

### 4.5. Cell Viability Assay

The MTT (3-(4,5-dimethylthiazol-2-yl)-2,5-diphenyltetrazolium bromide) assay was used to measure the cellular metabolic activity of PANC-1 cells as an indicator of cell viability, proliferation, and cytotoxicity. Cells were cultured at a density of 2 × 10^5^ cells/well in 12-well plates. After exposure to DIO-NPs, the cells were rinsed with phosphate-buffered saline (PBS) and 500 μL of 1 mg/mL yellow tetrazolium salt MTT solution (prepared in PBS) was added to each well. The plates were incubated for 2 h at 37 °C allowing the formation of intracellular purple formazan by metabolically active cells. The formazan crystals were solubilized with 250 μL/well of isopropanol and the absorbance was read at 595 nm. Cell viability was calculated in percentages as a ratio of treated cells vs. untreated cells (corresponding to 100%).

### 4.6. Iron Uptake Quantification

#### 4.6.1. DAB-Enhanced Perls’ Prussian Blue Staining

This is one of the most sensitive tests for iron detection. The Fe^3+^ ions present in sample react with the ferrocyanide and result in the formation of a bright blue pigment (Prussian blue). Thus, the PANC-1 cells were seeded on coverslips inside a 12-well plate filled with culture medium at a density of 2 × 10^5^ cells/well. After exposure to DIO-NPs, cells were fixed with paraformaldehyde (PFA) for 10 min and then washed twice with PBS and twice with a 0.3% Triton solution. Next, the cells were incubated for 15 min with 1% H_2_O_2_ to covert Fe^2+^ into Fe^3+^ ions. Following other three washes with PBS, the cells were incubated for 30 min with Perl’s solution (5% potassium hexacyanoferrate (II) trihydrate, 31254, Honeywell, MN, USA: 5% HCl, 1:1). After three washes with distilled water for 10 min each, a 0.5 mg/mL 3,3′-diaminobenzidine tetrahydrochloride (DAB) solution (D5637, Sigma, St, Louis, MO, USA) was added for 10 min following a 30 min incubation with a 0.5 mg/mL DAB solution containing 0.033% H_2_O_2_. The slides were washed again three times with distilled water and stained with 1% Luxol Fast Blue (L0294, Sigma, St, Louis, MO, USA) for one min. The slides were washed with 95% alcohol for 10 s, with butanol for 10 s, and with toluene twice for 10 s. Finally, a drop of acrytol mounting medium (3801720, Leica, Wetzlar, Germany) was used to mount the microscope slides, and images of cells were acquired using a DM2000 Leica microscope (Leica Microsystems, Groot Bijgaarden, Belgium) equipped with a Leica DFC 290 camera.

#### 4.6.2. Pearl’s Prussian Blue Colorimetry

A number of 2 × 10^6^ treated cells were digested in 100 µL of 5 N HCl solution on a water bath maintained at 80 °C for 4 h. Then, cells were centrifuged at 3000 rpm for 3 min to remove the digested cell debris. In a 96-well plate, 100 µL of cell digest was homogenized with 100 µL of 5% potassium hexacyanoferrate (II) trihydrate (ferrocyanide), and after 15 min the absorbance was read at 630 nm using a Stat Fax 4200 microplate reader (Awareness Technology, FL, USA). To quantify iron content, a calibration curve was obtained from IONP solutions ranging from 0 to 200 mM.

#### 4.6.3. Relaxometry Technique

The method is based on the intrinsic property of iron to increase the water proton relaxation rate (R_1_) by a value that is correlated to its concentration in a solution, cell suspension, or tissue [67]. Thus, a number of 2 × 10^6^ treated cells were digested by adding 100 µL of 5 N HCl solution and incubated for 4 h in a water bath at 80 °C. T1 (longitudinal) relaxation time was measured for the digested cell samples at 37 °C and 60 MHz using a 1.4 T Bruker Minispec mq60 (Brucker, Karlsruhe, Germany). The R_1_ value was calculated based on the reciprocal relation between the longitudinal relaxation rate of iron R_1_ and T_1_ relaxation time: R_1_(s) = 1/T_1_(s^−1^). Finally, R_1_ values of digested PANC-1 cells in the absence of nanoparticles (R_1_^diamagnetic^) were subtracted from the R_1_ values of digested samples containing iron, and the resulted values were transformed into iron concentration using a calibration curve (0.01–1 mM iron) made in the same conditions.

### 4.7. Lysosomal Labeling

The distribution and density of lysosomes in PANC-1 cells after exposure to DIO-NPs were analyzed by fluorescent microscopy using Lysotracker green DND-26 compound (L7526, Invitrogen, Carlsbad, USA). Briefly, cells were grown on coverslips inside a 12-well plate filled with culture medium (2 × 10^5^ cells/well). After treatment, the medium was removed and the cells were incubated in prewarmed (37 °C) medium with 100 nM Lysotracker green DND-26 for 30 min. Then, the cells were washed with PBS and fixed for 10 min with 4% PFA. Following two washes with PBS, the coverslips were mounted on microscope slides. Lysosomes were visualized in green fluorescence (ex. 504 nm/em. 511 nm) using an Olympus IX73 inverted microscope (Olympus, Tokyo, Japan) equipped with a Hamamatsu ORCA-03 camera. The density of lysosomes was analyzed by ImageJ software (version 1.47q, NIH, Bethesda, MD, USA). The fluorescence intensity of labeled lysosomes was correlated with the cell number from each picture and expressed as percentage of treated cells compared to control.

### 4.8. MRI Analyses

For MRI analyses, 2 × 10^6^ cells exposed to DIO-NPs were mixed with 100 μL of 2% gelatin and placed into 7 mm diameter NMR tubes at −20 °C to solidify. MRI scan was performed on a 7 T MRI scanner including a Bruker Biospec imaging system equipped with a Brucker PharmaScan horizontal magnet (Bruker Instruments, Ettlingen, Germany). Transversal sections over the tubes were acquired using a 4.5 cm^2^ field of view (FOV) and 1 mm slice thickness. The T_2_-weighted images were obtained with a multi-slice multi-echo (MSME) imaging sequence using the Bruker Paravision 5.1 software (Bruker Instruments, Ettlingen, Germany).

### 4.9. Toxicity Assessment

#### 4.9.1. ROS Production

Dichlorofluorescein-di-acetate (H_2_DCF-DA) was used as a fluorescent probe for the detection of ROS generation. Briefly, 2 × 10^5^ PANC-1 cells were seeded into Petri dishes and cultured overnight. After 24, 48, and 72 h of DIO-NP exposure, 10 μM H_2_DCF-DA (D6883, Sigma) was added in the culture medium and cells were incubated at 37 °C for 30 min. After two washes with PBS, the cells were collected and the fluorescence was determined at 488/525 nm (excitation/emission) using a Jasco FP-750 spectrofluorometer (Jasco Inc, Tokyo, Japan). In parallel, cells from each sample were counted using a Burker-Türk chamber. Results were calculated as relative fluorescent units (RFU) per number of cells and expressed in percentages of sample vs. control.

#### 4.9.2. Intracellular GSH Content

Cell lysate from each sample was obtained after sonication on ice 3 times for 30 s with an ultrasonic UP50H processor (Hielscher Ultrasound Technology, Teltow, Germany) and then deproteinized with 5% sulfosalicylic acid (1:1). The GSH concentration was assessed according to supplier’s instructions provided with Glutathione Assay Kit (CS0260, Sigma-Aldrich, Darmstadt, Germany). A calibration curve (3.125–50 μM) was prepared using reduced glutathione solution as standard. The results were expressed in nmoles GSH/mg of protein and represented in percentages of sample vs. control.

#### 4.9.3. Lipid Peroxidation

The level of malondialdehyde (MDA), a product of lipid peroxidation, was assessed by a method described by Dinischiotu et al., (2013) [68]. Thus, a volume of 200 µL of sample properly diluted was mixed with 0.1 N HCl and incubated for 20 min at room temperature. Then, 0.025 M thiobarbituric acid (TBA) was added and the reaction mix was incubated for 65 min at 37 °C. The fluorescence of MDA was recorded at a 520/549 nm (excitation/emission). A calibration curve (0.05–5 μM) prepared from 1,1,3,3-tetramethoxypropane standard solution was used to calculate the MDA concentration. The results were calculated as nmoles of MDA/mg of protein and represented in percentages of sample vs. control.

#### 4.9.4. LDH Assay

Measurement of LDH leakage in the culture medium serves as a potential marker of cell injury and death. LDH activity was determined using the method described by Vanderlinde (1985) [69] based on the LDH-catalyzed reaction of conversion of pyruvate into lactate. Briefly, a volume of 30 µL cell culture medium was mixed with 900 µL of 0.63 mM pyruvate solution and 15 µL of 13 mM NADH solution to start the reaction. During 5 min, the decrease in NADH absorbance after its oxidation was recorded each 10 s in a PerkinElmer spectrophotometer at 340 nm. LDH activity was calculated from the slope of the absorbance curve. The activity was calculated in units per mg protein (U/mg) where 1 U represents 1 µmol substrate hydrolyzed per minute, and finally data were expressed in percentages related to control. The protein content expressed as mg/mL was determined using Bradford’s method with bovine serum albumin as standard [70].

#### 4.9.5. Caspase-1 Activity

The activity of caspase-1 that recognizes the sequence YVAD was assessed using Caspase-1/ICE Colorimetric Protease Assay Kit (K111, BioVision, Milpitas, CA, USA) according to supplier’s instructions. Chromophore *p*-nitroanilide (*p*NA) was spectrophotometrically detected at 405 nm after cleavage from the labeled substrate YVAD-*p*NA. The fold increase in caspase-1 activity was calculated by comparing the absorbance of *p*NA from treated sample with the one from untreated cells and the results were expressed in percentages of sample vs. control.

### 4.10. Western Blot Analysis

The cell lysate mixed with loading buffer containing 0.25 M Tris-HCl, pH 6.8, 15% SDS, 50% glycerol, 25% β-mercaptoethanol, and 0.01% bromophenol blue were heated for 5 min at 95 °C. A quantity of 40 µg proteins from each sample was loaded onto SDS-PAGE polyacrylamide gels (10%) using the mini-cell system Mini Protean 3 from Bio-Rad. The electrophoretic separation was performed using a Tris-Gly (0.05 M) running buffer with 0.1% SDS at 70–90 V. Separated proteins were then transferred at 350 mA for 90 min onto polyvinylidene difluoride (PVDF) membrane using a wet-blot transfer system (Bio-Rad, Hercules, CA, USA) and a standard transfer buffer (pH 8.3) containing 25 mM Tris, 192 mM glycine, and 20% methanol. After transfer, blot membranes were exposed overnight to monoclonal anti-Hsp60 (sc-13115, Santa Cruz, Dallas, TX, USA), monoclonal anti-Hsp70 (sc-32239, Santa Cruz, Dallas, TX, USA), monoclonal anti-Hsp90 (ab13492, Abcam Inc., Cambridge, MA, USA), polyclonal anti-p53 (sc-6243, Santa Cruz, Dallas, TX, USA), and monoclonal anti-β-actin (A1978, Sigma-Aldrich, St. Louis, MO, USA) primary antibody solutions. Revelation of proteins bands was performed according to supplier’s instructions using the Western Breeze Chromogenic Anti-Mouse and Anti-Rabbit kits (WB7103, WB7105, Invitrogen, Carlsbad, CA, USA). β-actin was used as reference. The immunoreactive bands were visualized and captured with a ChemiDoc Imaging System (Bio-Rad, Hercules, CA, USA) and quantified with ImageLab 6.1 software (Bio-Rad Hercules, CA, USA). All values of samples were referred to the respective β-actin values.

### 4.11. Statistical Analysis

Data were obtained from three independent experiments and expressed as percentages of mean ± standard deviation (SD). Statistical analysis was performed in GraphPad Prism version 8 using two-way ANOVA with the Geisser–Greenhouse correction followed by Dunnett’s multiple comparison test. * *p* < 0.05, ** *p* < 0.01, and *** *p* < 0.001 were considered statistically significant comparing sample vs. control.

## 5. Conclusions

This study provides evidence of new dextran-covered maghemite nanoparticles with improved cell internalization capacity and enhanced MR contrast in pancreatic cancer cells, also revealing the potential toxicity mechanism that can be induced at high doses (56 μg/mL) of these NPs. Of significant interest, the present study showed that the accumulation of the DIO-NPs is correlated with MR contrast and toxicity levels, providing the dosage of biocompatible conditions that assure a safe utilization as theranostic platforms. We found also that DIO-NPs may cause necrotic cell death of pancreatic cancer cells, probably as a result of an enhanced lysosomal degradation, an antitumor effect that might be exploited for theranostic uses. Our study also sheds light on the roles of HSPs and p53 proteins in the protection of pancreatic cancer cells after exposure to DIO-NPs, contributing to the understanding of regulatory and adaptive mechanisms against oxidative stress.

## Figures and Tables

**Figure 1 ijms-24-03307-f001:**
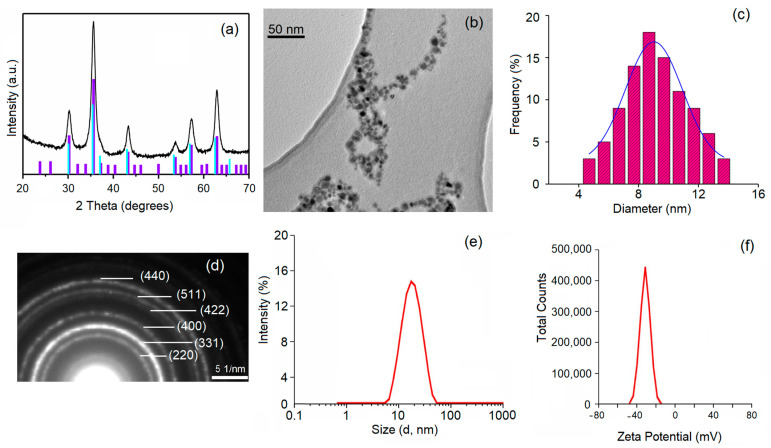
Characteristics of DIO-NPs. (**a**) X-ray powder diffraction patterns and reference patterns for maghemite (ICSD-PDF No. 1346; violet) and magnetite (ICSD-PDF No. 629; cyan); (**b**) TEM micrograph; (**c**) TEM histogram; (**d**) selected area electron diffraction (SAED); (**e**) DLS hydrodynamic size distribution of DIO-NPs in aqueous suspension and (**f**) zeta potential.

**Figure 2 ijms-24-03307-f002:**
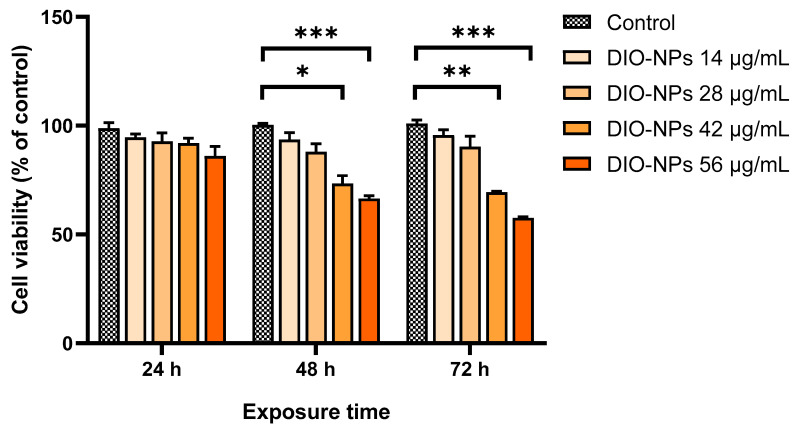
Cell viability of PANC-1 cells after exposure to different concentrations (14, 28, 42, and 56 μg/mL) of DIO-NPs for 24, 48, and 72 h. Data (*n* = 3) are expressed as percentages related to control ± SD. The results were considered statistically significant at * *p* < 0.05, ** *p* < 0.01, and *** *p* < 0.001 (sample vs. control).

**Figure 3 ijms-24-03307-f003:**
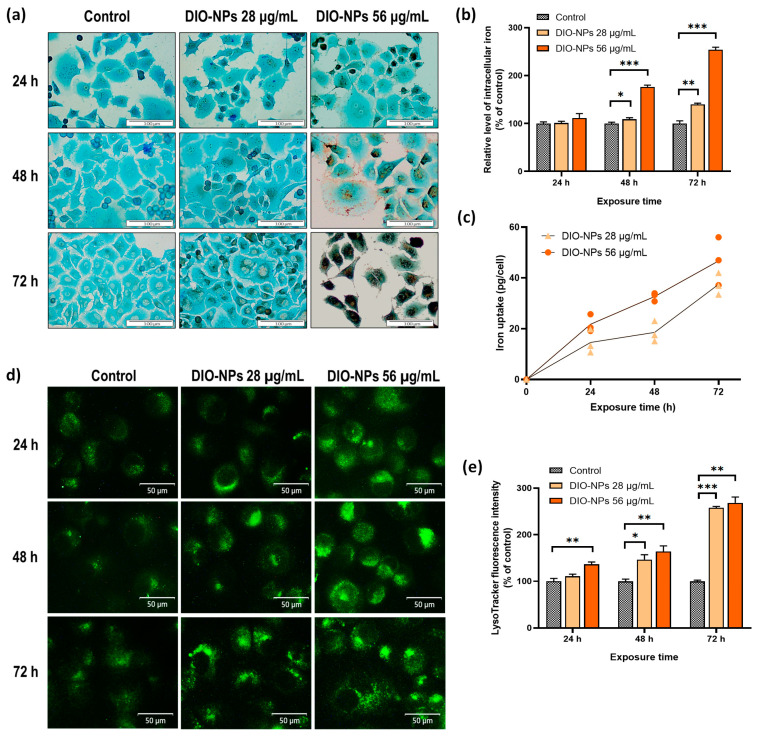
Cellular uptake of DIO-NPs in PANC-1 cells after 24, 48, and 72 h of exposure. (**a**) Representative images of DAB-enhanced Prussian blue staining of iron deposits (brown color) accumulated in the cytoplasm of PANC-1 cells (blue color, LuxolFast Blue). Scale bars, 100 μm; (**b**) intracellular iron content quantified by colorimetric Perls’ reaction; (**c**) cell NP-loading estimated by relaxometry (pg iron/cell). Each symbol represents an independent biological replicate; (**d**) representative images with the subcellular localization of lysosomes in PANC-1 cells using LysoTracker Green DND-26. Scale bars, 50 μm; (**e**) quantification of lysosome density by measuring the LysoTracker Green DND-26 fluorescence intensity. Data (*n* = 3) from (**b**) and (**d**) are expressed as percentages related to control ± SD. The results were considered statistically significant at * *p* < 0.05, ** *p* < 0.01, and *** *p* < 0.001 (sample vs. control).

**Figure 4 ijms-24-03307-f004:**
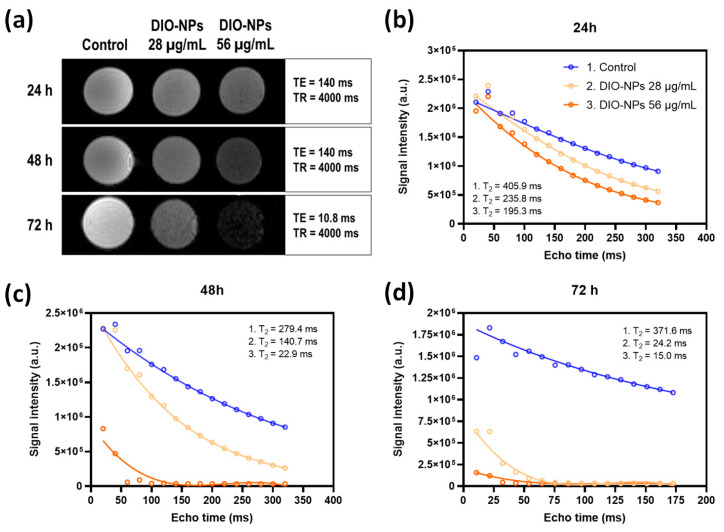
MR image contrast of DIO-NPs in PANC-1 cells after 24, 48, and 72 h of exposure. (**a**) T2-weighted images of PANC-1 control cells and cells treated with DIO-NPs measured with a 7 T MRI scanner; the graphs represent the curve fitting of signal intensity as a function of echo time for the estimation of transverse relaxation time T2 of DIO-NPs at a concentration of 0 (blue line), 28 (light orange line), and 56 μg/mL (dark orange line) after (**b**) 24, (**c**) 48, and (**d**) 72 h.

**Figure 5 ijms-24-03307-f005:**
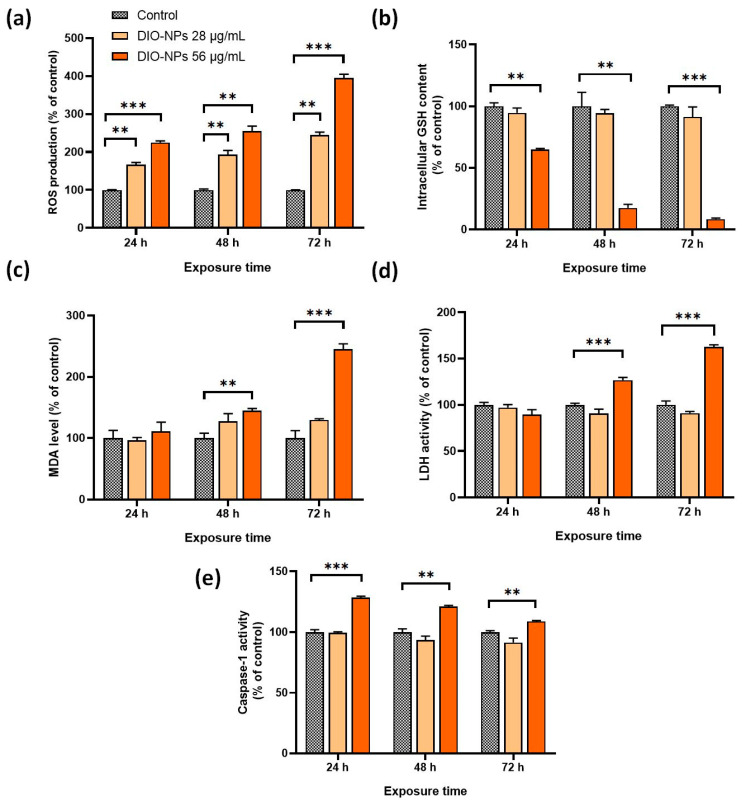
Toxic effects induced by DIO-NPs (28 and 56 μg/mL) in PANC-1 cells after 24, 48, and 72 h of exposure. (**a**) Measurement of intracellular ROS level by 2,7-DCF-DA staining; (**b**) spectrophotometric determination of intracellular GSH content; (**c**) fluorescent quantification of MDA as lipid peroxidation products; (**d**) quantification of LDH released in culture medium as a result of membrane damage; (**e**) estimation of caspase-1 activity by spectrophotometric detection of the chromophore p-nitroanilide (pNA) after cleavage from the labeled substrate YVAD-pNA. Data (*n* = 3) are expressed as percentages related to control ± SD. The results were considered statistically significant at ** *p* < 0.01 and *** *p* < 0.001 (sample vs. control).

**Figure 6 ijms-24-03307-f006:**
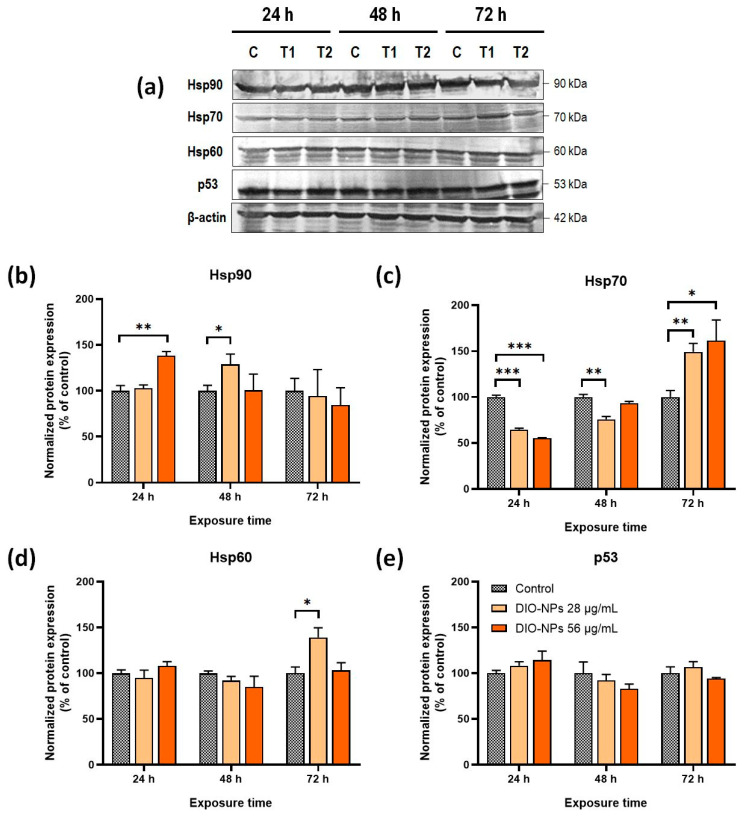
Heat shock proteins and p53 protein expression in PANC-1 cells after exposure to DIO-NPs (28 and 56 μg/mL) for 24, 48, and 72 h. (**a**) Protein bands of Hsp90, Hsp70, Hsp60, p53, and β-actin on blot images; bar plots representing the quantification of protein levels of (**b**) p53; (**c**) Hsp60; (**d**) Hsp70; (**e**) Hsp90 with ImageJ software. All values were normalized to β-actin. Data (*n* = 3) are expressed as percentages related to control ± SD. The results were considered statistically significant at * *p* < 0.05, ** *p* < 0.01, and *** *p* < 0.001 (sample vs. control). Abbreviations: C = control cells, T1 = cells treated with 28 μg/mL DIO-NPs, T2 = cells treated with 56 μg/mL DIO-NPs.

## Data Availability

Not applicable.

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
