# Peer review of "New Insights into the Biological Response Triggered by Dextran-Coated Maghemite Nanoparticles in Pancreatic Cancer Cells and Their Potential for Theranostic Applications"

_ijms, 2023, doi:10.3390/ijms24043307_

Round 1

Reviewer 1 Report

The current work focuses on the biological response triggered by dextran-coated maghemite nanoparticles in pancreatic cancer cells and their potential for theranostic applications. The author’s great effort into the improved manuscript but minor issues should be addressed. 

Abstract

Iron oxide-dextran for Theranostic application has been intensively studied. So, first, show the novelty of this work and then the main outcomes.

Keywords

Why no Keywords related to the application: Theranostic, diagnosis, therapy

Introduction

The introduction is providing sufficient background, and the most relevant references are included, but the novelty of this work is not highlighted and the author's contribution was unclear compared to other previous works. 

Materials and Methods 

write details on the synthesis of dextran-coated ɣ-Fe2O3 nanoparticles in the current work

What is the condition analysis for XRD, and scanning rate?

Results 

- Why claimed maghemite nanoparticles, it could be magnetite nanoparticles, 

In Reference 53, which you use to support your claim, they prepared magnetite. Also, Reference 57, which you use to support your claim, used (ICSD card no. 01-083-0112) but in your work claimed by (ICSD-PDF No. 79196).

XRD should interpretation on more detail to cover this point with real citations for supporting

- Indexed peaks should be inserted in figure XRD to compare

- Line 142, “The rings observed were in good accordance with the XRD pattern.” This claim needs more details in the text to support it or put it in SAED figure directly

Discussion

- Line 327, “The core of maghemite was covered with dextran to improve its dispersibility” before this sentence should insert the limitation as

e.g. “The agglomeration of magnetic particles is due to the presence of high surface energy between the prepared magnetic nanoparticles and the presence of magnetic dipole-dipole interactions.” DOI: 10.1080/01932691.2016.1140584

- Line 331 -333, “The colloidal stability of the DIO-NP suspension was indicated by the zeta potential value of -31.7 mV”, after this results value, it should insert details as

e.g. “ Nanoparticles’ stability is very important for biomedical applications to achieve expectable and consistent outcomes [26]. The low value of zeta potential of MNPs implies that the nanoparticle may show poor stability in aqueous solutions. Low zeta potential values will improve Van der Waals interparticle attractions and causes rapid coagulation and flocculation of nanoparticles. On the other hand, the higher value of zeta potentials implies that the nanoparticle may show good stability in aqueous solutions. There is a specific zeta potential value (≈ ± 30 mV) that determines the stability of nanoparticles. At this value, high electrostatic repulsive forces between the nanoparticles occur.” need a citation

- Therapy application is not focused and highlighted, also what about cell selectivity, localizing in the infected cells only to avoid effect on healthy cell

- Why not compared the results with commercial iron oxide-coated dextran as Resovist

Author Response

Responses for Reviewer 1

The current work focuses on the biological response triggered by dextran-coated maghemite nanoparticles in pancreatic cancer cells and their potential for theranostic applications. The author’s great effort into the improved manuscript but minor issues should be addressed.

We thank to the reviewer for the attention granted to our paper. We greatly appreciated all the remarks and comments that certainly contributed to the improvement of our manuscript.

Abstract

  • Iron oxide-dextran for Theranostic application has been intensively studied. So, first, show the novelty of this work and then the main outcomes.

Response: We thank for this comment. The novelty of this study is the characterization of maghemite nanoparticles’ biological response at a low dose versus a high dose in PANC-1 cells in order to address the potential use of these NPs in the theranostic application of pancreatic cancer. Our results provide the dosage of biocompatible and toxic conditions in pancreatic cancer cells, degree of cellular uptake, MR contrast, and the mechanism of toxicity as well as roles of HSPs and p53 proteins in the biological response of pancreatic cancer cells to DIO-NPs.

Several changes have been made in the manuscript. Please see page 1-abstract and page 3 lines 119-120.

Keywords

  • Why no Keywords related to the application: Theranostic, diagnosis, therapy

Response: We thank for this comment. The suggested words are a great idea that could increase the visibility of our paper. So, we added the word “theranostic” to the Keywords.

Introduction

  • The introduction is providing sufficient background, and the most relevant references are included, but the novelty of this work is not highlighted and the author's contribution was unclear compared to other previous works.

Response: We thank the reviewer for this comment. The novelty of our study resides in the characterization of maghemite nanoparticles’ biological response at a low-dose versus a high-dose in pancreatic cancer cells. Compared with previous studies, in this paper, the toxicity mechanism at low-dose vs. high-dose was revealed, as well as the contribution of several parameters to the toxicity mechanisms of dextran-coated IONPs on pancreatic cancer cells. For example, expression of heat shock proteins (Hsp60, Hsp70, Hsp90), caspase-1 enzyme activity, and p53 expression.

In our previous research, we showed the biological effects of DIO-NPs on lymphocytes (reference 43 in the manuscript - Balas, M.; Ciobanu, C.S.; Burtea, C.; Stan, M.S.; Bezirtzoglou, E.; Predoi, D.; Dinischiotu, A. Synthesis, Characterization, and Toxicity Evaluation of Dextran-Coated Iron Oxide Nanoparticles. Metals 2017, 7, 63). Here, we addressed pancreatic cancer because this is one of the highly aggressive types of cancer facing the lack of detection tools for its diagnosis in the early stages, and efficient treatments. The toxicity profile of maghemite nanoparticles is less presented in the scientific literature compared to magnetite nanoparticles as they are considered highly biocompatible nanoparticles. Even at non-toxic doses of maghemite nanoparticles, depending on different factors (size, shape, structure, solubility, concentration, surface modification, and cell/tissue type) changes in cellular parameters may occur. A deep understanding of these might guide the theranostic applications for effective outcomes.

Materials and Methods

  • write details on the synthesis of dextran-coated É£-Fe2O3 nanoparticles in the current work
  • What is the condition analysis for XRD, and scanning rate?

Response: We thank the reviewer very much for these remarks. In agreement with the reviewer’s suggestion, the synthesis process was detailed and the conditions for XRD were specified.

Results

  • Why claimed maghemite nanoparticles, it could be magnetite nanoparticles,
  • In Reference 53, which you use to support your claim, they prepared magnetite. Also, Reference 57, which you use to support your claim, used (ICSD card no. 01-083-0112) but in your work claimed by (ICSD-PDF No. 79196).
  • XRD should interpretation on more detail to cover this point with real citations for supporting
  • Indexed peaks should be inserted in figure XRD to compare
  • Line 142, “The rings observed were in good accordance with the XRD pattern.” This claim needs more details in the text to support it or put it in SAED figure directly

Response: In agreement with the suggestions of reviewer, clarifications were made in the text regarding the obtained material. Moreover, the XRD analysis was presented in more detail. Figure 1a was changed in the manuscript.  The reference patterns for maghemite (ICSD-PDF No. 1346; violet) and magnetite (ICSD-PDF No. 629; cyan) have been added. On the other hand, the SAED pattern of the synthesized DIO-NP was indexed and Figure 1d was changed.

Discussion

  • Line 327, “The core of maghemite was covered with dextran to improve its dispersibility” before this sentence should insert the limitation as

e.g. “The agglomeration of magnetic particles is due to the presence of high surface energy between the prepared magnetic nanoparticles and the presence of magnetic dipole-dipole interactions.” DOI: 10.1080/01932691.2016.1140584

Response: We thank very much for his competent observations! Comments and references have been added in agreement with the reviewer´s suggestions.

  • Line 331 -333, “The colloidal stability of the DIO-NP suspension was indicated by the zeta potential value of -31.7 mV”, after this results value, it should insert details as

e.g. “ Nanoparticles’ stability is very important for biomedical applications to achieve expectable and consistent outcomes [26]. The low value of zeta potential of MNPs implies that the nanoparticle may show poor stability in aqueous solutions. Low zeta potential values will improve Van der Waals interparticle attractions and causes rapid coagulation and flocculation of nanoparticles. On the other hand, the higher value of zeta potentials implies that the nanoparticle may show good stability in aqueous solutions. There is a specific zeta potential value (≈ ± 30 mV) that determines the stability of nanoparticles. At this value, high electrostatic repulsive forces between the nanoparticles occur.” need a citation

Response: Following the reviewer’s observation additional information and references in agreement with the colloidal stability were added to the discussions.

  • Therapy application is not focused and highlighted, also what about cell selectivity, localizing in the infected cells only to avoid effect on healthy cell

Response: We agree with the reviewer’s comment. Indeed, the therapeutic application of DIO-NPs was not the focus of our paper, but the biological response of pancreatic cancer cells to our NPs. The therapeutic applications of IONPs, which are well known, were mentioned in the introduction section. For example, IONPs generate heat through interaction with low-frequency electromagnetic radiation (hyperthermia). This heat can destroy cancer cells without affecting the healthy cells around them. More, IONPs can be functionalized by targeting moieties such as antibodies or peptides to selectively deliver drugs or therapeutic agents to specific malignant tissues without harming healthy cells. Additionally, the toxicity of DIO-NPs through ROS generation can be exploited for the eradication of cancer cells by delivering them directly to the tumor. All these are also potential therapeutic applications for DIO-NPs addressed in our paper.

  • Why not compared the results with commercial iron oxide-coated dextran as Resovist

Response: We thank to reviewer for this comment. It would be very interesting to compare our results with those obtained for Resovist on pancreatic cancer cells. In comparison with our NPs, Resovist (Ferucarbotran) consists in a mixture of magnetite and maghemite covered with carboxydextran, with a particle size of about 60 nm. According to previous reports, the problems faced with Resovist use, include: poor cellular uptake and failure to provide a high-contrast signal (Singh, Abhalaxmi; Dilnawaz, Fahima; Mewar, Sujeet; Sharma, Uma; Jagannathan, N. R.; Sahoo, Sanjeeb Kumar (2011). Composite Polymeric Magnetic Nanoparticles for Co-Delivery of Hydrophobic and Hydrophilic Anticancer Drugs and MRI Imaging for Cancer Therapy. , 3(3), 842–856). In our study, we obtained a high contrast effect due to a high T2 relaxivity and a high cellular uptake of DIO-NPs in PANC-1 cells but we did not find information about Resovist for the same parameters to compare the results. What biological response is triggered by the Resovist nanoparticles in PANC-1 cells and what contrast signal will be achieved is not known and remain to be elucidated.

Reviewer 2 Report

The authors present dextrane coated iron oxide nanoparticles (synthesized in-house) as promising agents for theranostics. As an example, the authors used pancreatic cancer cells. Introduction extensively outlines the significance of theranostics approach to deal with pancreatic cancer in a comprehensive way.

Comments:

Line 55-57 could be a part of the paragraph above.

Line 124 such statement is recommended to support with the reference.

Line 129, 134, 139 etc - carefully check that all the special symbols are inserted correctly like the error for the size etc

Fig 1 

- scale bars and axes titles should be bigger.  

- Fig.1(a) simulated patterns for iron oxide (magnetite and maghemite) plotted together with the data obtained by the authors are beneficial to observe the peaks positions.

- Fig.1d - why the scale bar demonstrated as “5 1/nm”?

- it's recommended to combine Fig.1(c) and Fig.1(e) to qualitatively see the shift of the size distribution

Fig 2

- Question: a considerable decrease in the cells viability is seen after 48h exposure and at least 42ug/ml. How much time does reticuloendothelial system (RES) need to identify and filter out dextrane coated IONPs if we are talking about in vivo? If it is under 48h - then it can be concluded that these dextran coated IONPs are ineffective in causing PANC-1 cells any harm. Please, elaborate on this question.

Conclusions requires concrete results, e.g. Line 673 "....at high doses..." - doses should be stated.

The Reviewer thanks the authors for a well structured quality manuscript. After addressing comments and suggestions to improve the manuscript, it is recommended for further publishing process.

Author Response

Responses for Reviewer 2

The authors present dextrane coated iron oxide nanoparticles (synthesized in-house) as promising agents for theranostics. As an example, the authors used pancreatic cancer cells. Introduction extensively outlines the significance of theranostics approach to deal with pancreatic cancer in a comprehensive way.

Comments:

  • Line 55-57 could be a part of the paragraph above.

Response: the indicated paragraph (lines 55-57) has been moved above.

  • Line 124 such statement is recommended to support with the reference.
  • Line 129, 134, 139 etc - carefully check that all the special symbols are inserted correctly like the error for the size etc

Response: We thank the reviewer very much for these competent remarks. Considering the reviewer's comments, the related changes were made in the text.

  • Fig 1 
    • scale bars and axes titles should be bigger.  
    • 1(a) simulated patterns for iron oxide (magnetite and maghemite) plotted together with the data obtained by the authors are beneficial to observe the peaks positions.
    • 1d - why the scale bar demonstrated as “5 1/nm”?
    • it's recommended to combine Fig.1(c) and Fig.1(e) to qualitatively see the shift of the size distribution

Response: Following the reviewer’s observation, Figure 1 has been changed according to the referent's requirements.

To respond to the reviewer’s question, this is the scale provided by the instrument’s software, which is accurate given the fact that the diffraction pattern is a two-dimensional projection of the reciprocal crystal lattice, therefore, in a simple TEM image the scale bar shows the length scale in the real space whereas similar information obtained from the diffraction pattern measurements are in a reciprocal space.

  • Fig 2
    • Question: a considerable decrease in the cells viability is seen after 48h exposure and at least 42ug/ml. How much time does reticuloendothelial system (RES) need to identify and filter out dextrane coated IONPs if we are talking about in vivo? If it is under 48h - then it can be concluded that these dextran coated IONPs are ineffective in causing PANC-1 cells any harm. Please, elaborate on this question.

Response: The reviewer rises a very good question. According to previous studies, the time necessary for the RES to identify and filter out dextran-coated IONPs in vivo can vary depending on several factors, including size, concentration, surface charge, etc. Typically, several hours to a few days are necessary for RES to clear IONPs from the bloodstream. For example, dextran-coated iron oxide nanoparticles such as Ferumoxtran-10 composed of smaller nanoparticles (dH = 15–50 nm) is characterized by a much longer circulation time (human blood half-life between 24 and 36 h) than Ferumoxides with dH of 62–80 nm ( with a human blood half-life  between 3.9 and 8 min )(Nowak-Jary J, Machnicka B. Pharmacokinetics of magnetic iron oxide nanoparticles for medical applications. J Nanobiotechnology. 2022 Jun 27;20(1):305.). Taking into account this, we can assume that our nanoparticles with a dH = 16.3 nm may possibly have a longer circulation time probably between 24 h and 36 h. More, a high negative value (for example -35 mV) of the surface minimizes the tendency to agglomeration of nanoparticles and thereby the tendency to absorb plasma proteins providing prolonged circulation time to an extreme degree (Ghorbani M, Bigdeli B, Jalili-baleh L, Baharifar H, Akrami M, Dehghani S, et al. Curcumin-lipoic acid conjugate as a promising anticancer agent on the surface of gold–iron oxide nanocomposites: A pH-sensitive targeted drug delivery system for brain cancer theranostics. Eur J Pharm Sci. 2018;114:175–88.). In the case of our nanoparticles, we registered a surface charge of -31.7 mV which might contribute to their persistence.

Although the cytotoxicity of dextran IONPs was observed after 48 h we showed that these NPs could initiate an inflammatory response in PANC-1 cells at a dose of 56 μg/mL in the first 24 as a toxicity mechanism which proves that PANC-1 cells may suffer damage in the first 24 h. However, further research is needed to determine an exact time frame for the clearance of IONPs by RES in vivo. Regarding this comment, a phrase was added also in the manuscript on Page 12.

  • Conclusions requires concrete results, e.g. Line 673 "....at high doses..." - doses should be stated.

Response: We agree with the reviewer’s comment. The dose was specified in the manuscript. Please see page 18 of the manuscript.

  • The Reviewer thanks the authors for a well structured quality manuscript. After addressing comments and suggestions to improve the manuscript, it is recommended for further publishing process.

We thank the reviewer for the attention granted to our paper. We greatly appreciated all the remarks and comments that certainly contributed to the improvement of our manuscript.